# Study on the Dynamic Splitting Mechanical Properties of Annular Sandstone Specimens with Temperature–Water Coupling in a Coal Mine

**Qi Ping** [1,2,3,*] , **Qi Gao** [2,3], **Yulin Wu** [2,3], **Chen Wang** [2,3], **Kaifan Shen** [2,3], **Shuo Wang** [2,3], **Shiwei Wu** [2,3] **and Yijie Xu** [2,3]

1   State Key Laboratory of Mining Response and Disaster Prevention and Control in Deep Coal Mine, Anhui University of Science and Technology, Huainan 232001, China
2   Engineering Research Center of Mine Underground Projects, Ministry of Education, Anhui University of Science and Technology, Huainan 232001, China; 2020200277@aust.edu.cn (Q.G.); 2020200404@aust.edu.cn (Y.W.); 2020200327@aust.edu.cn (C.W.); 2020200341@aust.edu.cn (K.S.); 2020200372@aust.edu.cn (S.W.); 2021200374@aust.edu.cn (S.W.); 2021200476@aust.edu.cn (Y.X.)
3   School of Civil Engineering and Architecture, Anhui University of Science and Technology, Huainan 232001, China
*   Correspondence: ahpingqi@163.com or qping@aust.edu.cn; Tel.: +86-139-5645-9398

**Featured Application: The basic physical parameters and dynamic properties of annular and intact sandstone specimens treated by temperature–water coupling are compared and analyzed.**

**Abstract:** With the gradual deepening of mine excavation depth, the strong disturbance of deep strata becomes more and more obvious. Rock's failure under blasting mainly depends on its dynamic tensile strength. The changes in rock's dynamic properties are obviously affected by temperature and water. In order to study the dynamic tensile properties of annular sandstone specimens under the influence of temperature and water, deep sandstone was drilled, followed by water bath tests at eight temperatures (25~95 °C). It can be seen from the analysis of test results that the mass and volume growth rates of the annular and the intact sandstone specimens first increased and then decreased, while the density growth rate first decreased and then increased. The mass and volume growth rates of the annular sandstone specimens were smaller, but the density growth rate was larger. Because of the increase in water temperature, the dynamic compressive strength first increased and then decreased. The dynamic tensile strength of the annular sandstone specimen was lower. The average strain rate and peak strain also showed a quadratic function relationship of first decreasing and then increasing with the increase in water temperature. The average strain rate of the annular sandstone specimen was smaller, but the peak value changed greatly. The Brazilian disc validity condition is applicable to two failure conditions of sandstone specimens. Through XRD and SEM analysis, we found that the changes in the dynamic properties of sandstone specimens were not due to their own material composition, but to the damage to their structure caused by the temperature–water coupling effect.

**Keywords:** rock impact dynamics; sandstone annular specimen; SHPB; Brazilian disc split test; temperature–water coupling

---





## 1. Introduction

With the continuous increase in the human demand for mineral resources, shallow resources are not enough to meet human demand, so the depth of mine excavation is gradually increasing. However, with the increase in depth, the excavation becomes more difficult, and the underground strata environment becomes more complicated. Therefore, it is necessary to study the mechanical properties of ore at the bottom of mines, and how they are affected by various strong disturbances.

Temperature and groundwater are common influencing factors in the process of mine excavation. Ping [1] applied temperature–water coupling treatment to sandstone, and carried out dynamic compression tests to analyze the dynamic properties of sandstone, such as peak strength, peak strain, average strain rate, and elastic modulus. Ping [2] conducted dynamic splitting tests on the temperature–water coupling of sandstone with different impact pressures, and found that the dynamic properties were affected by strain rate. Wang et al. [3] found that the dynamic properties of sandstone are affected by water bonding force and the Stefan effect. Roy [4] treated different types of sandstone for different saturation times, and concluded that their mechanical properties and fracture toughness decreased with the increase in saturation. Wang et al. [5] investigated the mechanical properties and failure law of two kinds of coal rock with different moisture content. Teng et al. [6] researched the degradation mechanism of shale under different water-bearing states. Deng et al. [7] considered the influence of five different water contents on the splitting tensile strength of layered sandstone. Wang et al. [8] conducted splitting tensile strength tests of sandstone Brazilian discs under different temperatures and water contents, providing basic reference data for the standardization of splitting strength and engineering applications. Ping et al. [9,10] conducted SHPB impact tests on sandstone and limestone at different temperatures to analyze their dynamic properties. It was concluded that different effects of high temperature made the peak strength of the rock first increase, and then decreases. Zhang [11] used an SHPB device to study the dynamic properties and damage characteristics of sandstone at different temperatures. Qin et al. [12] analyzed the influence of mesoscopic changes on the uniaxial compressive strength of high-temperature sandstone via acoustic damage, X-ray diffraction, and scanning electron microscopy. Xu et al. [13] combined and predicted the rate-dependent effect of dynamic tensile strength of two igneous rocks by using incubation time as a criterion. Ke et al. [14,15] studied the dynamic and static mechanical properties of rocks under freeze–thaw cycles. Zheng et al. [16] enriched the detection methods of water-induced mine disasters by studying the influence of water on the induction signals of rock failure charge. Chang et al. [17] studied the tensile strength characteristics of sandstone, mudstone, coal, and medium sandstone under different water content states. Geng [18] tested dynamic damage to sandstones using acoustic emission technology, and concluded that water absorption reduces brittleness but increases ductility. Taiki [19] carried out uniaxial compression tests on clay with cracks of different prefabricated lengths and different water contents, and concluded that the deformation modulus depends on the water content, but has nothing to do with crack length. Li [20] studied the energy storage and acoustic emission characteristics of rock under different water content states. Wang [21] studied the influence of moisture content on the characteristic strength of soft coal, and found that the shear strength reached the maximum when the moisture content was 4.23%. Zhao [22] studied the surface characteristics, porosity, and permeability of limestone treated with different temperatures by dry-ice cooling. Liu [23] conducted SHPB tests on limestone treated at different temperatures under different confining pressures, and the test results showed that 400 °C was an inflection point of dynamic properties.

The strength and other mechanical properties of deep sandstone are affected not only by temperature and groundwater, but also by its own inhomogeneity and size [24]. Scholars at home and abroad have studied the mechanical properties and failure modes of annular specimens with holes by simulating their own cracks and internal channels. Wang et al. [25] carried out an experimental study on radial compression of annular granite under various temperature and humidity conditions, and found that the maximum tensile strain could be used as a failure judgment parameter for annular granite. Wu et al. [26] used annular sandstone specimens with different inner diameter to conduct Brazilian disc splitting tests. Yang et al. [27] constructed a discrete element model for Brazilian disc splitting testing of a single-hole disc, and studied the influence of aperture and eccentricity on specimen deformation and strength characteristics. You et al. [28] conducted triaxial compression tests on marble samples with two different channels to analyze their bearing capacity and deformation characteristics. It was concluded that the different characteristics of marble affect the deformation and damage to marble specimens. Ying [29] conducted impact loading tests

on a tunnel's surrounding rock in different directions to analyze its fracture toughness and other characteristics, which was of great value to tunnel engineering research.

It can be seen that the rock dynamic properties of temperature–water coupling are worthy of further study. Since the rock's inhomogeneity is also the main factor affecting its mechanical properties, this test drilled the sandstone specimen and then carried out temperature–water coupling treatment at different temperatures (25 °C~95 °C), so as to study its physical properties, dynamic properties, and failure mode. Annular sandstone specimens can be used not only to simulate the influence of inhomogeneity on mechanical properties, but also to simulate the failure law of shaft walls affected by transverse load in the process of shaft excavation. Because rock's failure mainly depends on its tensile strength, but it is difficult to directly measure the tensile strength, the Brazilian disc splitting test was adopted in this experiment [30].

## 2. Materials and Methods

### 2.1. Sandstone Specimen Processing and Preparation

According to the requirements of the test specification [31,32] and the dynamic test specification [33], the sandstone required for the test was cored, cut, and polished. The height and diameter of the processed sandstone specimen were 25 mm and 50 mm, respectively. In the preliminary experiment, the failure characteristics and data processing of the 10 mm aperture specimen were more representative, which was convenient for the final results' analysis. Therefore, a 10 mm drill bit was used for drilling in this experiment. The center of the circle was found on the specimen and marked with a marker. Then, a drilling machine was used for drilling. In order to prevent the influence of friction heating on the strength of the drill bit, water was used to cool and lubricate the drill bit during drilling. The effect diagram of drilling the specimen is shown in Figure 1.

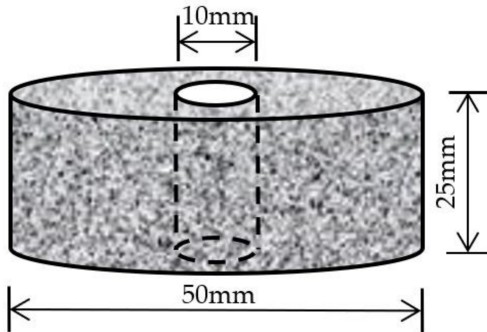

**Figure 1.** Drilling effect drawing.

### 2.2. Experimental Plan

The sandstone in this paper was taken from the roadway sandstone of a coal mine, and the depth of the mine's bottom is about 600 m. Combined with the local geological and climatic conditions, we improved the test scheme. Because the temperature of the bottom of the mine is kept above 25 °C all year round, and the water temperature reaches 100 °C, boiling occurs. The annular and intact sandstone specimens were subjected to temperature–water coupling treatment at eight temperatures (25 °C~95 °C), followed by SHPB impact testing. It was found that when the impact pressure was 0.3 MPa, the waveform obtained from the test was better and more conducive to data processing. Therefore, the impact pressure was set to 0.3 MPa in this test. At the end of the test, the test data were processed, and the dynamic properties and failure modes of the annular sandstone specimens were analyzed and compared with those of the intact sandstone specimens, so as to analyze the influence rule of the annular sandstone's dynamic properties.

After processing, the two specimens were treated with thermostatic water baths at different temperatures. In order to ensure that the sandstone specimens met the water saturation condition, each temperature gradient was treated with a 48h constant-temperature

water bath. After the temperature–water coupling effect, the surface of the sandstone specimen was dried, its mass and its internal and external diameter were measured, and then the specimen was placed on the SHPB pressure bar for dynamic splitting tensile testing after different water bath temperatures.

### 2.3. SHPB Test Device

The pressure rod device (SHPB) used in this study is shown in Figure 2.

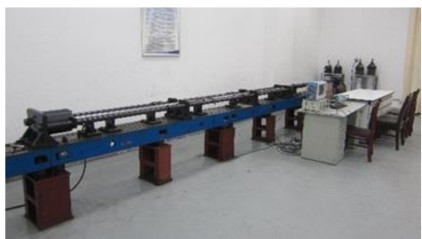

**Figure 2.** SHPB test device.

Brazilian disc splitting tests were carried out on the annular and intact sandstone specimens treated by temperature–water coupling with eight temperature gradients. Immediately after the impact, the test data were saved and the post-impact fragments were collected. The clamping mode of the specimen is shown in Figure 3.

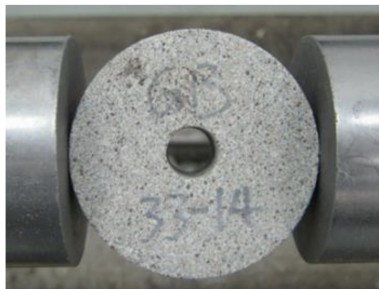

**Figure 3.** Clamping method of the specimen.

Through the data collected with the signal collection device, the data were calculated and processed, and the dynamic parameters required by the test—such as impact load $P(t)$, dynamic strain $\varepsilon_s$, and average strain rate $\dot{\varepsilon}_s$—were obtained. The calculation principle [34,35] is shown in Formula (1).

$$\left. \begin{array}{l} P(t) = E_0 A_0 [\varepsilon_I(t) - \varepsilon_R(t)] = E_0 A_0 \varepsilon_T(t) \\ \varepsilon(t) = \frac{2C_0}{D} \int_0^\tau [\varepsilon_I(t) - \varepsilon_T(t)] dt = \frac{2C_0}{D} \int_0^\tau \varepsilon_R(t) dt \\ \dot{\varepsilon}(t) = -\frac{2C_0}{D} [\varepsilon_I(t) - \varepsilon_T(t)] = -\frac{2C_0}{D} \varepsilon_R(t) \end{array} \right\} \tag{1}$$

where $E_0$ is the elastic modulus, and $A_0$ is the cross-sectional area of the press bar material.

$\varepsilon_I$ is the incident strain, $\varepsilon_R$ is the reflected strain, and $\varepsilon_T$ is the projected strain.

$C_0$ is the longitudinal wave velocity of the pressure bar device;

$\rho_0$ is the material density of the pressure rod device;

$D$ is the diameter of the sandstone specimen;

$\tau$ is the duration of the stress waves.

## 3. Analysis of Test Data

### 3.1. Physical Properties Analysis

Photos of sandstone specimens treated with different water temperatures are shown in Figure 4

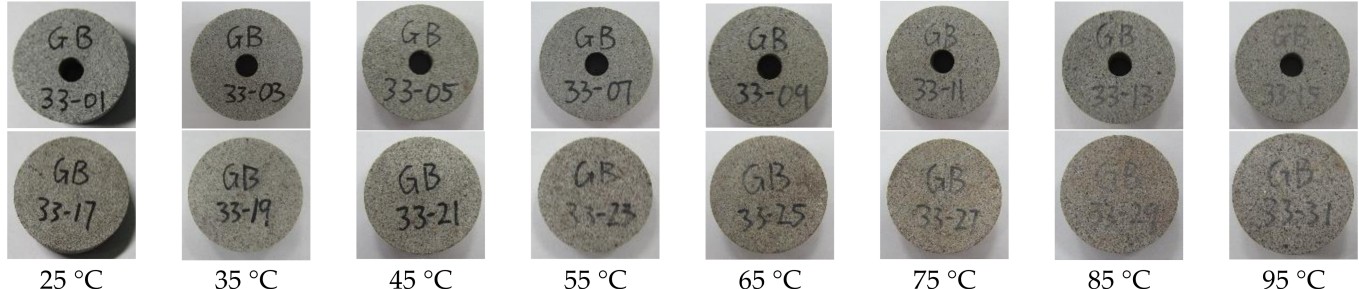

| 25 °C | 35 °C | 45 °C | 55 °C | 65 °C | 75 °C | 85 °C | 95 °C |

**Figure 4.** The apparent morphology of sandstone specimens after the effect of temperature–water coupling.

As can be seen from the picture, the increase in the water temperature lightens the surface color of the specimen. The specimen numbers also gradually became blurred under the temperature–water coupling effect. When touching the specimens' surface gently with the hand, the higher the water temperature, the stronger the surface frosting feeling, and the more granular fine sand floating on the specimen surface.

Before and after the temperature–water coupling effect, the physical basic data of the sandstone specimens were calculated, and the results are shown in Table 1.

**Table 1.** Comparison of physical data before and after temperature–water coupling.

| Temperature (°C) | Specimen Number | Before Temperature–Water Coupling | | | After Temperature–Water Coupling | | |
| | | Mass (g) | Volume (mm$^3$) | Density (g/mm$^3$) | Mass (g) | Volume (mm$^3$) | Density (g/mm$^3$) |
|---|---|---|---|---|---|---|---|
| 25 | GB33-01 | 122.98 | 47.05 | 2.61 | 123.20 | 47.10 | 2.62 |
| | GB33-02 | 123.25 | 47.24 | 2.61 | 123.63 | 47.28 | 2.62 |
| 35 | GB33-03 | 123.81 | 47.51 | 2.61 | 124.12 | 47.54 | 2.61 |
| | GB33-04 | 123.58 | 47.09 | 2.62 | 123.93 | 47.17 | 2.63 |
| 45 | GB33-05 | 122.31 | 47.23 | 2.59 | 122.61 | 47.30 | 2.59 |
| | GB33-06 | 123.51 | 47.14 | 2.62 | 123.89 | 47.18 | 2.63 |
| 55 | GB33-07 | 124.03 | 47.30 | 2.62 | 124.37 | 47.30 | 2.63 |
| | GB33-08 | 121.11 | 46.91 | 2.58 | 121.44 | 47.01 | 2.58 |
| 65 | GB33-09 | 121.45 | 47.42 | 2.56 | 121.83 | 47.48 | 2.57 |
| | GB33-10 | 123.96 | 47.31 | 2.62 | 124.24 | 47.33 | 2.62 |
| 75 | GB33-11 | 123.76 | 47.41 | 2.61 | 124.10 | 47.44 | 2.62 |
| | GB33-12 | 124.21 | 47.51 | 2.61 | 124.52 | 47.56 | 2.62 |
| 85 | GB33-13 | 124.13 | 47.61 | 2.61 | 124.41 | 47.62 | 2.61 |
| | GB33-14 | 124.12 | 47.60 | 2.61 | 124.45 | 47.62 | 2.61 |
| 95 | GB33-15 | 122.34 | 47.29 | 2.59 | 122.61 | 47.29 | 2.59 |
| | GB33-16 | 121.98 | 47.11 | 2.59 | 122.22 | 47.12 | 2.59 |
| 25 | GB33-17 | 133.23 | 48.85 | 2.73 | 133.73 | 49.07 | 2.73 |
| | GB33-18 | 126.55 | 48.80 | 2.59 | 127.32 | 49.16 | 2.59 |
| 35 | GB33-19 | 127.25 | 48.89 | 2.60 | 127.91 | 49.21 | 2.60 |
| | GB33-20 | 125.96 | 49.00 | 2.57 | 126.60 | 49.39 | 2.56 |
| 45 | GB33-21 | 129.60 | 49.45 | 2.62 | 130.13 | 49.42 | 2.63 |
| | GB33-22 | 127.84 | 49.54 | 2.58 | 128.76 | 49.31 | 2.61 |
| 55 | GB33-23 | 126.77 | 48.94 | 2.59 | 127.31 | 49.00 | 2.60 |
| | GB33-24 | 129.52 | 49.65 | 2.61 | 130.40 | 50.37 | 2.59 |
| 65 | GB33-25 | 129.27 | 48.56 | 2.66 | 129.74 | 48.95 | 2.65 |
| | GB33-26 | 134.65 | 49.26 | 2.73 | 135.55 | 49.55 | 2.74 |
| 75 | GB33-27 | 125.92 | 48.89 | 2.58 | 126.49 | 49.14 | 2.57 |
| | GB33-28 | 127.46 | 48.92 | 2.61 | 128.12 | 49.16 | 2.61 |
| 85 | GB33-29 | 129.17 | 49.29 | 2.62 | 129.83 | 49.49 | 2.62 |
| | GB33-30 | 129.15 | 48.93 | 2.64 | 129.60 | 49.05 | 2.64 |
| 95 | GB33-31 | 127.22 | 48.85 | 2.60 | 127.72 | 48.87 | 2.61 |
| | GB33-32 | 128.66 | 49.04 | 2.62 | 129.09 | 49.05 | 2.63 |

The basic physical properties of sandstone before and after temperature–water coupling were analyzed according to the data in the table.

The curve of the mass growth rate is shown in Figure 5.

$$
\left.\begin{aligned}
m_1' &= 0.166 + 0.004t - 3.857 \times 10^{-5}t^2 \left(R^2 = 0.9731\right) \\
m_2' &= 0.308 + 0.010t - 9.538 \times 10^{-5}t^2 \left(R^2 = 0.9780\right)
\end{aligned}\right\} \tag{2}
$$

where $m_1'$ is the annular sandstone specimen's mass growth; $m_2'$ is the intact sandstone specimen's mass growth.

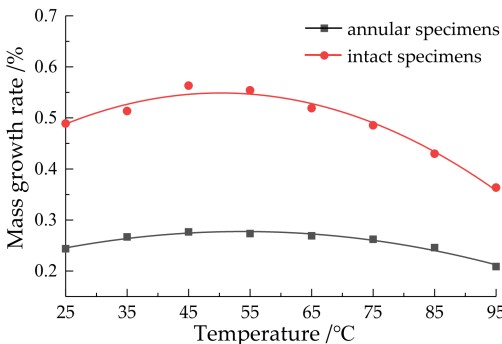

**Figure 5.** Mass growth rate varies with water temperature.

The mass growth rate curve satisfies the quadratic function relation, and an inflection point appeared at 45 °C. The mass growth rate of the intact sandstone specimens was larger. The correlation coefficients of the two specimens' fitting curves were 0.9731 and 0.9780, respectively, showing a strong correlation. Analysis of the reasons shows that the overall volume of the intact sandstone specimen was larger, and more water could be absorbed, which entered the internal cracks of the sandstone and increased its mass. At the same time, the increase in water temperature enlarged the cracks in the sandstone specimen, enabling it to absorb more water, so its mass increased first. However, with the increase in water temperature, some granular sand appeared and flaked off on the surface of the specimen, resulting in a decrease in quality. When the temperature exceeded 45 °C, the amount of degradation damage exceeded the amount of water absorption, so the mass growth rate decreased; thus, the mass growth rate first increased and then decreased.

The curve of the volume growth rate is shown in Figure 6.

$$
\left.\begin{aligned}
V_1' &= 0.040 + 0.003t - 4.032 \times 10^{-5}t^2 \left(R^2 = 0.9894\right) \\
V_2' &= 0.061 + 0.036t - 3.696 \times 10^{-5}t^2 \left(R^2 = 0.9860\right)
\end{aligned}\right\} \tag{3}
$$

where $V_1'$ is the annular sandstone specimen's volume growth; $V_2'$ is the intact sandstone specimen's volume growth.

The volume growth rate first increased and then decreased with the increase in water temperature, and the correlation coefficient reached 0.9894 and 0.9860, respectively, showing a strong correlation. The volume growth rate of the intact sandstone specimen was greater, because the volume of the intact sandstone specimen was larger and could absorb more water, resulting in the increase in internal cracking, so the volume growth rate was higher. Similarly, the increase in water temperature not only increased the volume, but also caused the deterioration and damage of the specimen, with part of the surface falling off, leading to a decrease in the overall volume, so the trend of first increasing and then decreasing appeared.

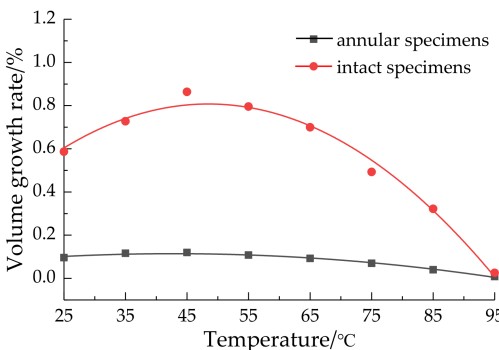

**Figure 6.** The volume growth rate varies with water temperature.

The density growth rate presents a law under mass and volume changes, as shown in Figure 7.

$$\left. \begin{array}{l} \rho_1' = 0.243 - 0.003t + 3.242 \times 10^{-5}t^2 \left( R^2 = 0.9515 \right) \\ \rho_2' = 0.362 - 0.025t + 2.649 \times 10^{-5}t^2 \left( R^2 = 0.9914 \right) \end{array} \right\} \tag{4}$$

where $\rho_1'$ is the annular sandstone specimen's density growth; $\rho_2'$ is the intact sandstone specimen's density growth.

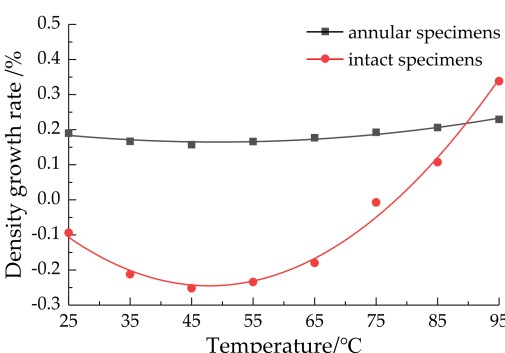

**Figure 7.** Density growth rate varies with water temperature.

Because volume growth is greater than mass growth, density first shows a decreasing trend. When the temperature exceeds 45 °C, the mass growth rate is greater than the volume growth rate, so the density growth rate tends to increase. Due to the better integrity and greater volume and mass of intact sandstone specimens, their density growth rate changes more obviously under the temperature–water coupling.

*3.2. Dynamic Performance Analysis*

When studying the dynamic properties of the two sandstone specimens, applying Vaseline to the surface of the specimen and pressure bar can reduce friction and end-face resistance, so as to improve the reliability of the test data. The impact test data have a large dispersion, so we prepared five sandstone specimens of each type. In order to make the test results more convincing, we selected two data within the error range of 15% and took their average value.

After processing the test data, the results are shown in Table 2.

**Table 2.** SHPB test data.

| Temperature (°C) | Specimen Number | Annular Sandstone Specimens | | | Specimen Number | Intact Sandstone Specimens | | |
|---|---|---|---|---|---|---|---|---|
| | | Peak Stress (MPa) | Average Strain Rate (s⁻¹) | Peak Strain (×10⁻³) | | Peak Stress (MPa) | Average Strain Rate (s⁻¹) | Peak Strain (×10⁻³) |
| 25 | GB33-01 | 17.06 | 84.50 | 2.86 | GB33-17 | 25.26 | 81.30 | 3.62 |
| | GB33-02 | 15.29 | 84.40 | 2.81 | GB33-18 | 26.19 | 81.70 | 3.54 |
| 35 | GB33-03 | 17.21 | 83.50 | 2.64 | GB33-19 | 26.91 | 80.80 | 3.43 |
| | GB33-04 | 16.69 | 83.30 | 2.73 | GB33-20 | 26.25 | 80.70 | 3.49 |
| 45 | GB33-05 | 16.48 | 83.60 | 2.50 | GB33-21 | 27.31 | 80.70 | 3.03 |
| | GB33-06 | 17.54 | 83.10 | 2.40 | GB33-22 | 26.60 | 79.90 | 3.38 |
| 55 | GB33-07 | 15.10 | 83.40 | 2.60 | GB33-23 | 26.62 | 80.30 | 3.58 |
| | GB33-08 | 17.52 | 83.80 | 2.88 | GB33-24 | 26.32 | 80.90 | 3.56 |
| 65 | GB33-09 | 16.73 | 84.60 | 3.24 | GB33-25 | 26.41 | 79.70 | 4.02 |
| | GB33-10 | 14.51 | 84.00 | 2.81 | GB33-26 | 26.10 | 83.20 | 4.09 |
| 75 | GB33-11 | 15.48 | 84.60 | 3.89 | GB33-27 | 25.73 | 82.00 | 4.52 |
| | GB33-12 | 15.08 | 85.30 | 3.97 | GB33-28 | 25.96 | 81.30 | 4.54 |
| 85 | GB33-13 | 15.00 | 85.90 | 4.93 | GB33-29 | 25.01 | 83.20 | 5.21 |
| | GB33-14 | 13.26 | 86.20 | 4.68 | GB33-30 | 25.33 | 83.10 | 5.25 |
| 95 | GB33-15 | 13.80 | 85.60 | 5.46 | GB33-31 | 21.97 | 85.20 | 5.61 |
| | GB33-16 | 12.95 | 88.00 | 5.02 | GB33-32 | 25.36 | 83.30 | 5.64 |

### 3.2.1. Stress–Strain Relationship Analysis

The stress–strain relationship of the two sandstone specimens is shown in Figure 8.

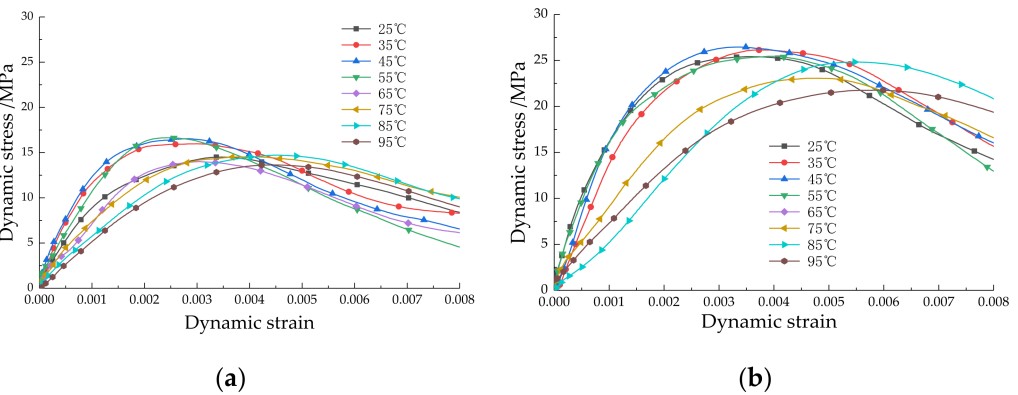

(**a**)            (**b**)

**Figure 8.** Stress–strain curves: (**a**) stress–strain curves of annular sandstone specimens; (**b**) stress–strain curves of intact sandstone specimens.

By comparing and analyzing the stress–strain curves of the two sandstone specimens with different structures, their dynamic properties can be preliminarily obtained. The dynamic tensile strength of annular sandstone is obviously lower than that of intact sandstone, due to the influence of pores. The slope of the curves of the two sandstone specimens also changes with the increase in temperature under the temperature–water coupling effect. The slope of the curves of the two sandstone specimens also changes with the increase in temperature under the temperature–water coupling effect.

### 3.2.2. Dynamic Tensile Strength Analysis

The tensile strength of the specimen is consistent with the peak stress, and the instability failure occurs when the sandstone specimen reaches the tensile strength. After testing and data processing, the peak of the dynamic stress–strain curve is the dynamic tensile strength.

Dynamic tensile strength is shown in Figure 9.

$$
\left.
\begin{aligned}
P_1(t) &= 15.045 - 0.085t + 0.001 \times 10^{-5}t^2 \left(\mathrm{R}^2 = 0.9598\right) \\
P_2(t) &= 23.133 - 0.145t + 0.001 \times 10^{-5}t^2 \left(\mathrm{R}^2 = 0.9682\right)
\end{aligned}
\right\}
\tag{5}
$$

where $P_1(t)$ is the annular sandstone specimen's dynamic tensile strength; $P_2(t)$ is the intact sandstone specimen's dynamic tensile strength.

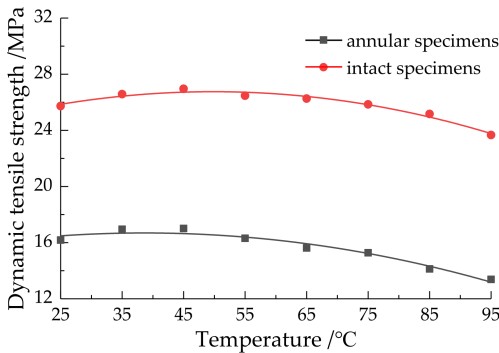

**Figure 9.** Relationship between dynamic tensile strength and water temperature.

According to the analysis, the peak tensile strength of the two sandstone specimens changed at 45 °C. Through quadratic function fitting, the correlation coefficients reached 0.9598 and 0.9682, respectively, showing strong correlation. In the process of testing, water entered the internal void. Considering the influence of water bonding force and the Stefan effect [36], the strength of the specimens was enhanced to a certain extent. After 45 °C, the effects of temperature and water on the damage to the specimen are greater than its enhancement. Partial spalling occurred on the surface of the specimens, internal cracks increased, and the specimens were more prone to instability failure. Therefore, the peak strength of both specimens gradually decreased. Because the annular sandstone specimen had lower integrity, its peak tensile strength was lower than that of the annular sandstone specimen.

### 3.2.3. Average Strain Rate Analysis

The failure difficulty of specimens can be reflected by the average strain rate.

The variation in the average strain rate with water temperature is shown in Figure 10.

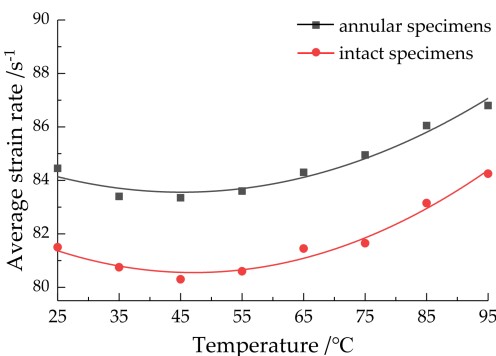

**Figure 10.** Relationship between average strain rate and water temperature.

Average strain rate first decreased but then increased with the change in water temperature, and an inflection point appeared at 45 °C. The correlation coefficients reached 0.9614 and 0.9750, respectively, by quadratic function fitting, indicating obvious correlation. This was related to the dynamic tensile strength. Under the influence of water bonding force and the Stefan effect, the dynamic tensile strength of the specimens increased, so the

failure difficulty of the specimens increased and the average strain rate decreased accordingly. When the water temperature exceeded the inflection point, the damaging effects of temperature and water were more obvious, so the specimens were easier to destroy and the average strain rate increased accordingly. Due to the influence of its structure, the dynamic tensile strength of the annular sandstone specimen was low, and this specimen was more prone to failure. Therefore, the average strain rate of the annular sandstone specimen was lower than that of the intact sandstone specimen.

$$\left.\begin{array}{l} \dot{\varepsilon}_1 = 86.454 - 0.128t + 0.001 \times 10^{-5}t^2 \ \left(R^2 = 0.9614\right) \\ \dot{\varepsilon}_2 = 84.258 - 0.157t + 0.002 \times 10^{-5}t^2 \ \left(R^2 = 0.9750\right) \end{array}\right\} \tag{6}$$

where $\dot{\varepsilon}_1$ is the annular sandstone specimen's average strain rate; $\dot{\varepsilon}_2$ is the intact sandstone specimen's average strain rate.

### 3.2.4. Peak Strain Analysis

The peak strain can also reflect the degree of failure and breakage to a certain extent.

The peak strain first decreased and then increased with the increase in temperature, and reached the inflection point at 45 °C, while the correlation coefficient reached 0.9713 and 0.9720, respectively, indicating an obvious correlation. Consistent with the dynamic tensile strength and average strain rate, the water bonding force and Stefan effect increased the specimens' strength, so the peak strain decreased. As the water temperature increased, damage was the main effect, making the specimens easier to damage and increasing the peak strain. According to the figure, the integrity, peak strength, and peak strain of intact sandstone specimens are higher than those of annular sandstone specimens. In the impact test, the annular hole played a buffer role, and the inner ring was squeezed first when it was impacted by the pressure bar, so the peak strain of the annular sandstone specimen was slightly lower than that of the intact sandstone specimen.

The variation of peak strain with water temperature is shown in Figure 11.

$$\left.\begin{array}{l} \varepsilon_{T1} = 4.101 - 0.076t + 9.554 \times 10^{-5}t^2 \ \left(R^2 = 0.9713\right) \\ \varepsilon_{T2} = 4.343 - 0.051t + 7.021 \times 10^{-5}t^2 \ \left(R^2 = 0.9720\right) \end{array}\right\} \tag{7}$$

where $\varepsilon_{T1}$ is the annular sandstone specimen's peak strain; $\varepsilon_{T2}$ is the intact sandstone specimen's peak strain.

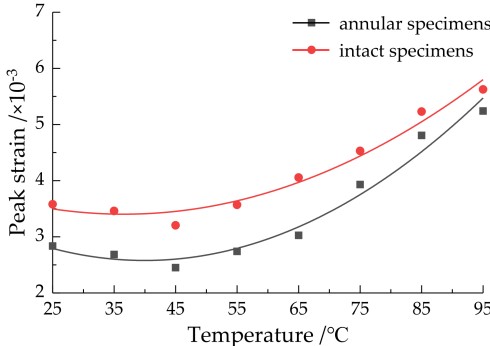

**Figure 11.** Relationship between peak strain and water temperature.

### 3.3. Failure Mode Analysis

Failure modes can also reflect the dynamic characteristics of sandstone specimens to a certain extent.

The failure modes of each water temperature gradient are shown in Figure 12.

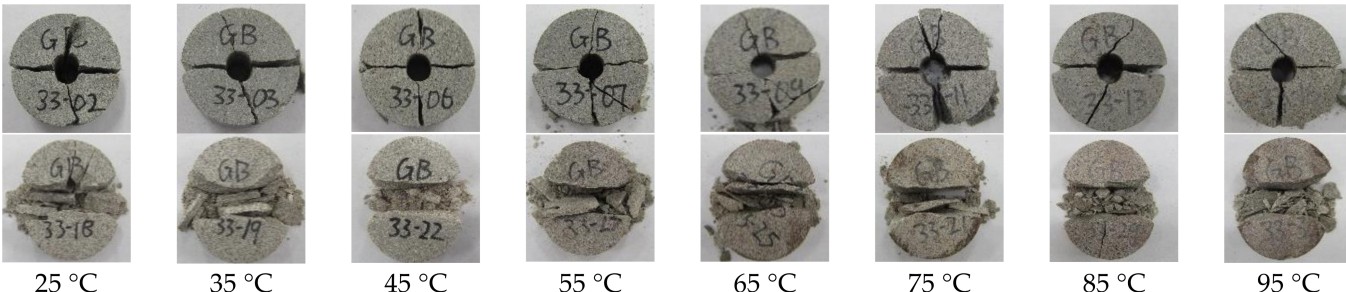

| 25 °C | 35 °C | 45 °C | 55 °C | 65 °C | 75 °C | 85 °C | 95 °C |

**Figure 12.** Fracture morphology of sandstone specimens.

Through observation and analysis, we found that the annular sandstone specimen under impact load was mainly divided into four parts, accompanied by some small fragments spalling. The specimen was first cracked along the axis of the incident bar, and then the upper and lower parts were extruded, cracked from the edge of the specimen, and destroyed along the direction of the internal fissure. The failure mode was type I failure. When the water temperature was 45 °C, the fragments were mainly divided into four parts of similar size, which were broken apart along the central axis and the vertical central axis. When the temperature was greater than or less than 45 °C, the strength of the specimen was low, and the size of fragments was not uniform. It can be concluded that the failure of the upper and lower parts of the specimen was mainly along the internal cracks, thus showing irregularity. Therefore, at 45 °C, the integrity of the sandstone specimens was better. The fracture morphology of the complete sandstone specimen was split along the central axis, leaving two fan-shaped fragments, small fragments, and powder. At the same time, we found that the fan-shaped area of the sandstone specimen was larger at 45 °C, indicating that its strength was higher. The failure modes of the two sandstone specimens meet the validity conditions of the Brazilian disc test.

The fracture morphology can be verified laterally, based on the dynamic tensile strength, average strain rate, and peak strain variation scale of the sandstone specimens. It was proven that 45 °C is an inflection point for the dynamic parameters of sandstone specimens with temperature–water coupling.

### 3.4. XRD Pattern and SEM Photo Analysis

After the impact test, an X-ray diffractometer and an electron microscope were used to conduct XRD and SEM tests on sandstone specimens, respectively, so as to measure their XRD patterns and SEM images. The measured data are shown in Figures 13 and 14, respectively.

After the analysis of the XRD patterns, no new substances were found in the sandstone after the temperature–water coupling effect. The composition of sandstone is mainly $SiO_2$, with a small amount of $K_2(PtCl)_4$ and $CaCuV_2O_7$. SEM images show the morphological characteristics of the sandstone fracture surface. As the temperature of the temperature–water coupling increased gradually, the fracture surface became coarser, and angular grains appeared at the fracture surface (the coarsest at 45 °C). The XRD patterns and SEM image analysis show that the dynamic properties of sandstone are mainly determined by the internal structural changes after the temperature–water coupling effect.

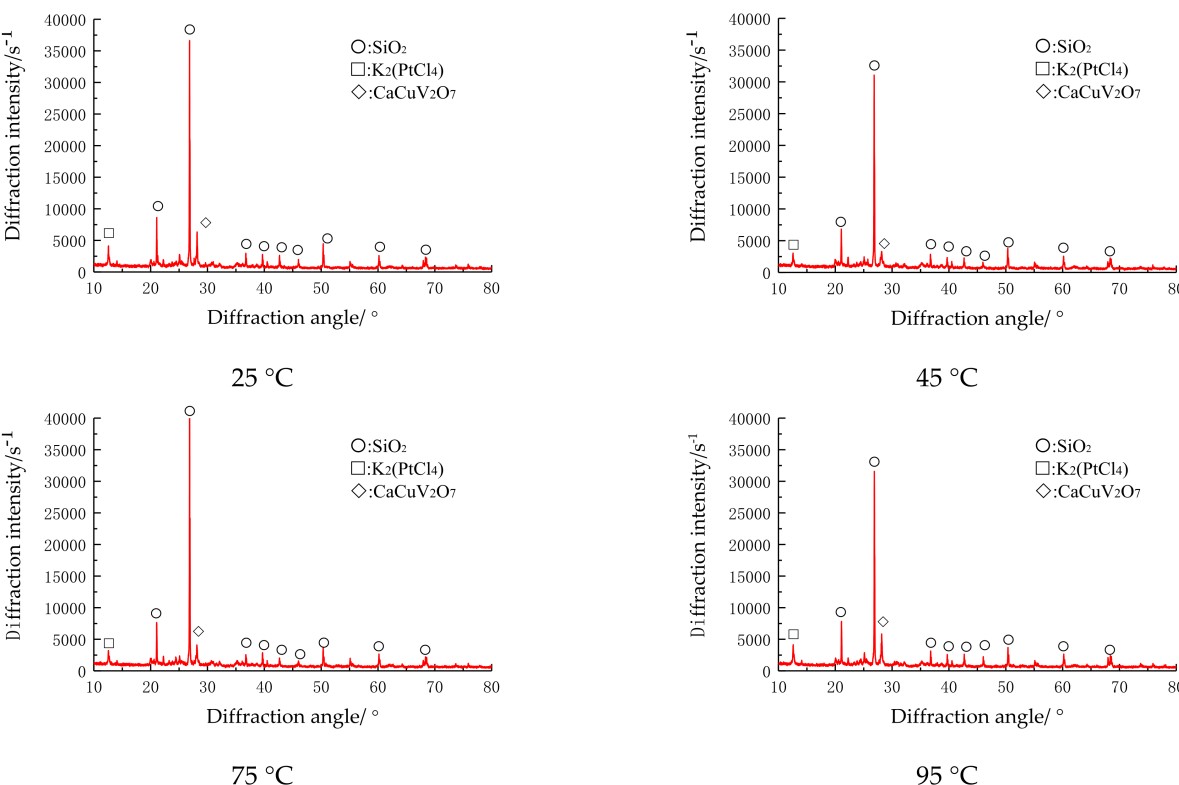

**Figure 13.** XRD patterns of sandstone specimens treated with different water bath temperatures.

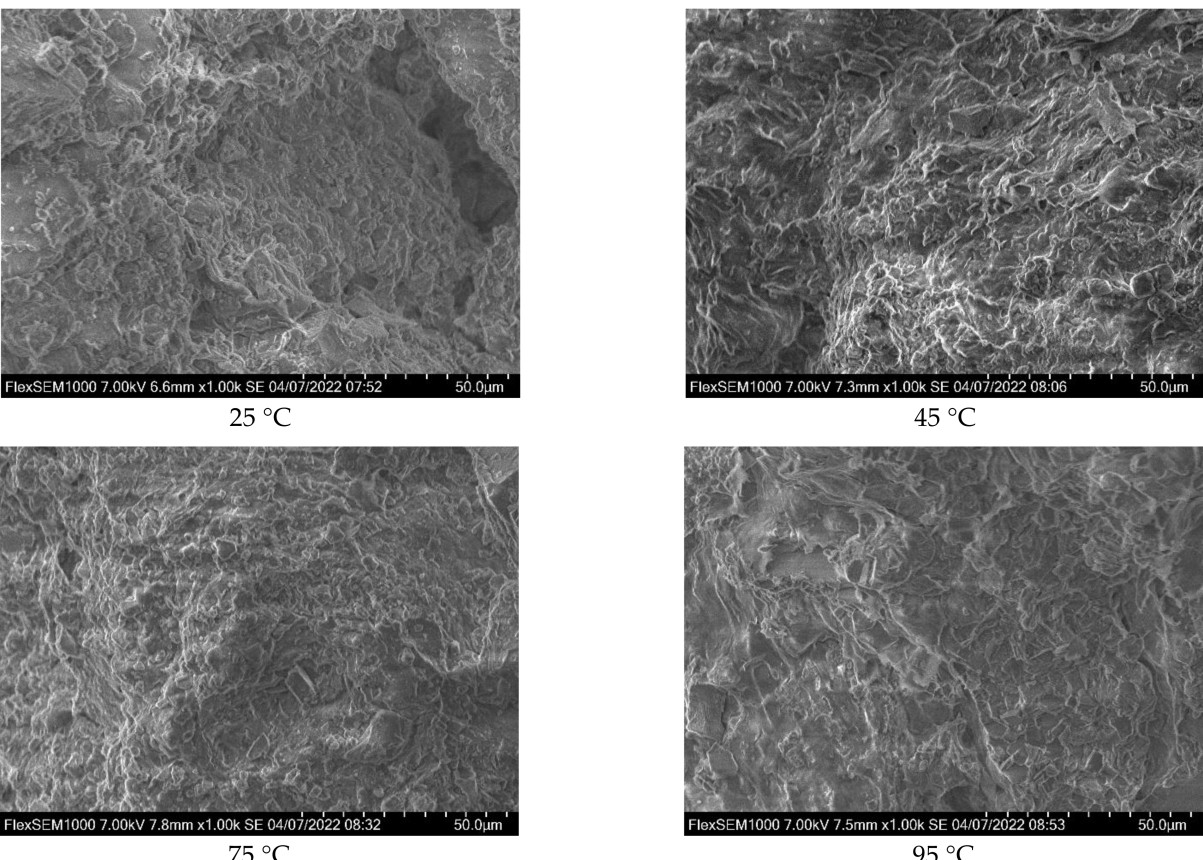

**Figure 14.** SEM photos of sandstone specimens treated by water bath at different temperatures.

## 4. Conclusions

In this study, eight temperature gradients (25 °C~95 °C) were used to conduct temperature–water coupling treatment for annular sandstone specimens and intact sandstone specimens, followed by testing of their dynamic splitting mechanical properties (SHPB). The following conclusions were obtained:

(1) After the water bath, the growth rate of the mass and volume of the sandstone specimens first increased and then decreased with the increase in temperature, and the growth rate of density first decreased and then increased under the joint influence of mass and volume changes, both of which met the quadratic function relationship.

(2) The peak strength, average strain rate, and peak stress of the two sandstone specimens all met the quadratic function relation, the correlation was very strong, and the law of increase and decrease at 45 °C changed.

(3) Due to the influence of its structure, the strength of annular sandstone is obviously lower than that of intact sandstone. However, the prefabricated holes in the annular sandstone specimens have a certain buffer effect on the impact load, so the strain is lower.

(4) The fracture morphology of the two sandstone specimens was consistent with the Brazilian disc splitting model. In addition, the fragmentation morphology changed with temperature to different degrees, and tended to be complete at 45 °C.

(5) The changes in the basic physical parameters and dynamic properties of sandstone samples after temperature–water coupling mainly depend on their internal structural damage, and their material composition does not change with the increase in the water bath temperature.

**Author Contributions:** Conceptualization, Q.P.; Data curation, Q.G.; Formal analysis, C.W.; Funding acquisition, Q.P.; Investigation, Q.G., Y.W. and Y.X.; Methodology, Q.P.; Project administration, S.W. (Shiwei Wu); Resources, Q.P. and K.S.; Software, K.S.; Supervision, Y.W.; Validation, S.W. (Shuo Wang); Writing—original draft, Q.G.; Writing—review & editing, Q.P. All authors have read and agreed to the published version of the manuscript.

**Funding:** This research was funded by the National Natural Science Foundation of China (no. 52074005, no. 52074006), Anhui University of Science and Technology Graduate Innovation Fund Project (no. 2021CX2032), and the National College Student Innovation and Entrepreneurship Training Program (no. 202110361022, no. 2021103661027, no.202110361032).

**Institutional Review Board Statement:** Not applicable.

**Informed Consent Statement:** Informed consent was obtained from all subjects involved in the study.

**Data Availability Statement:** The data used to support the findings of this study are available from the corresponding author upon request.

**Acknowledgments:** We would like to thank Anhui University of Science and Technology for providing the experimental conditions.

**Conflicts of Interest:** The authors declare no conflict of interest. The funders had no role in the design of the study; in the collection, analyses, or interpretation of data; in the writing of the manuscript, or in the decision to publish the results.

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
