# Peer review of "Study on the Dynamic Splitting Mechanical Properties of Annular Sandstone Specimens with Temperature–Water Coupling in a Coal Mine"

_applsci, doi:10.3390/app12094608_

Round 1
Reviewer 1 Report
The issues discussed in the publication are the most up-to-date.
However, I feel a bit unsatisfied with the research presented in the above article.
I miss a more detailed analysis of the material. What is the type of sandstone? EDS analysis together with SEM photos would significantly enrich the publications. They would also answer questions related to the influence of temperature and water and possible leaching of components from the sandstone structure.
The issues mentioned by the authors regarding the influence of temperature on sandstone leaching / degradation have not been exhausted. Have the authors investigated what percentage by mass has been washed out?
It would be good to examine the samples cooled down before the test, and then dry the samples after the tests and test the mass again.
The description for Figure 8 is more like the draft report, not the scientific analysis needed for the publication.
In the presented studies, I miss the statistics. How many samples of each type have been tested? What is the standard deviation of the presented results?
In point 2.2, the MPa unit was incorrectly marked.
In line 264 the unit should be in the same row as its value.
References has been largely oriented towards domestic publications. It would have to be expanded to include world publications. There is no DOI in the references part.
Author Response
请参阅附件

Reviewer 2 Report
The issues of testing rock under dynamic loads in the context of mine workings stability, is very interesting. I read the text with attention.
However, I have some comments:
- I lack petrographic information regarding the sandstone in question (age, formation, mineral composition, etc.). There is also a lack of information regarding its porosity, binder and other characteristics that will allow the reader to imagine how the rock in question might behave in studies. Also, nothing is known about the layering of the sandstone.
- I lack information about the condition of the rock mass at the time of sampling. How fractured are the rocks in question, are the grains in them not cracked, etc. This is also important in further considerations.
- the title of the paper should also be narrowed down to the location of the sandstone. Especially since the authors examine the sandstone from basically one location. This will firstly allow to state the facts of the analyses and secondly will enable the authors to publish similar papers with other sandstones and finally to synthesize their results.
4 I am also puzzled by the temperature range, as there are areas of permafrost where the temperature is negative and also areas of high geothermal. I would recommend using a wider temperature range.
5) Can we talk about repeatability of results for two sandstone samples?
6) Do the authors plan to study other sandstones and other rocks as well?
7) What is the influence of sandstone structure on the results?
The lack of discussion and the very laconic conclusions basically make it impossible to answer these questions.
In addition, only 26 literature items were cited, I think more can be found.
Dear authors, your work is large, I appreciate the results and tests carried out, however, I kindly ask you to reconsider my comments.
Round 2
Reviewer 1 Report
I would like to thank the authors for correcting the publication. Following the changes. I accept the paper for publication
Author Response
Thanks for the reviewer's careful review and high evaluation of this paper. We have revised and supplemented the article under the advice of experts, and the quality of the article has been improved. We have checked the article again and polished it in English. We hope the article can be accepted after this revision.
Reviewer 2 Report
Dear Authors, I sincerely thank you for the enormous amount of work you have put into this text. Thank you for considering my suggestions. I am very grateful to the authors for the clarifications and changes in the text, which in my opinion is now much improved.
I have only one more remark: In the petrographic description of the sandstones, the authors cited XRD data. It is not enough. One should ideally show a photo from a polarizing microscope. The examined sandstones may consist in 90% of SiO2 but it can be both dressed quartz, regenerated quartz, quartzite lithoclasts etc. Besides, it is the nature of the clasts that determines the strength of the sandstone, so it is worth showing the polarizing microscope image.
Please feel free to add to this content.
I have no further comments.
Author Response
Thanks for the reviewer's careful review and recognition of the content of the article. We have supplemented and modified the article according to your suggestions. We rechecked the article and polished it in English.
Because the university is located in Huainan city, Anhui Province, China, the campus has been closed due to the epidemic. Express delivery is also at a standstill, and we cannot supplement the polarized light test for the time being. Thank you very much for the advice of experts. We will pay attention to the analysis of experimental results by microscopic factors in future studies.
Thanks again for the expert's opinion, our article quality has been improved under your suggestion. I hope I can get hired smoothly after this revision.